# Sacubitril/valsartan preserves kidney function in rats with cardiorenal syndrome after myocardial infarction

Kaja Knudsen Bergo[1]*, Einar Sjaastad Nordén[1,2], Bård Andre Bendiksen[1,2], Emil Knut Stenersen Espe[1], Gary McGinley[1], Ida Marie Hauge-Iversen[1], Rizwan Iqbal Hussain[3], Sabine Leh[4,5], Hans-Peter Marti[4,6], Lili Zhang[1], Ivar Sjaastad[1], Alessandro Cataliotti[1]

1 Institute for Experimental Medical Research, Oslo University Hospital and University of Oslo, Oslo, Norway, 2 Oslo New University College, Oslo, Norway, 3 Agiana Pharmaceuticals AS, Oslo, Norway, 4 Department of Clinical Medicine, University of Bergen, Bergen, Norway, 5 Department of Pathology, Haukeland University Hospital, Bergen, Norway, 6 Department of Medicine, Haukeland University Hospital, Bergen, Norway

* kaja.bergo@gmail.com

## Abstract

Renal dysfunction in heart failure increases mortality, limits treatment options and blunts responses to therapy. Angiotensin receptor-blocker and neprilysin inhibitors (ARNI) may preserve renal function by modulating both the renin-angiotensin-aldosterone system and the natriuretic peptide system. We investigated the renal effects of the ARNI Sacubitril/valsartan (Sac/Val) in rats with systolic dysfunction secondary to myocardial infarction. Male Sprague-Dawley rats underwent surgical MI induction and were randomized to six weeks of treatment with vehicle, valsartan or Sac/Val, and compared to sham operated animals. Renal function was evaluated by creatinine clearance, mean arterial pressure (MAP) by tail-cuff measurements, and cardiac function by magnetic resonance imaging and echocardiography. Vehicle treated animals developed cardiorenal syndrome, with impaired cardiac systolic function and mild renal dysfunction. Both valsartan and Sac/Val preserved renal function compared to vehicle (creatinine clearance $mL/min$ [median with interquartile range]; sham 5.4 [4.8–6.0], vehicle 4.5 [4.1–5.1], valsartan 5.1 [5.1–5.5], Sac/Val 5.1 [5.0–5.6]; vehicle vs valsartan p=0.034 and vehicle vs Sac/Val p=0.044). MAP was reduced by both treatments compared to sham and vehicle groups (MAP $mmHg$; sham 131 [116–138], vehicle 123 [115–132], valsartan 108 [99–112], Sac/Val 111 [99–119]; sham vs valsartan p<0.001 and sham vs Sac/Val p=0.003, vehicle vs valsartan p=0.006 and vehicle vs Sac/Val p=0.041). Only Sac/Val reduced left atrial dilatation (diameter $mm$; sham 4.1 [3.7–4.4], vehicle 4.6 [3.8–5.6], valsartan 4.6 [4.1–5.5], Sac/Val 3.9 [3.6–4.5]; vehicle vs Sac/Val p=0.047, valsartan vs Sac/Val p=0.017) despite no improvement in systolic function in either treatment group. Sac/Val initiated in the acute post-MI phase preserved renal function to the same extent as valsartan alone and uniquely reduced left atrial dilatation, suggesting additional benefits beyond renoprotection in the setting of persistent systolic dysfunction.

**Data availability statement:** Data supporting this study are included within the article. The data set is deposited in and available at the zenodo.org repository, with DOI: 10.5281/zenodo.16756734.

**Funding:** Sources of funding: - Oslo University Hospital: providing salary for the first author KKB (no specific grant number) - The Blix Family Fund for the Promotion of Medical Research: 40 000 NOK (no specific grant number) to KKB - Olav Raagholt og Gerd Meidel Raagholts stiftelse: 40 000 NOK (no specific grant number) to KKB - Simon Fougner Hartmanns Familiefond: economic support for ultrasound machine (grant number 101539-461587) to IS - Novartis Pharmaceuticals Corporation provided the research group (IS) with Sacubitril/valsartan and Valsartan used in this study free of charge and without restrictions for use within the project. - Novartis Pharmaceuticals Corporation provided support in the form of salaries for author RIH (no specific grant number) at the time the study was conducted, but did not have any additional role in the study design, data collection and analysis, decision to publish, or preparation of the manuscript. Also, RIH primarily contributed to the present project after his employment at Novartis was ended. The specific role of the author is stated in the 'author contributions' section The funders had no role in study design, data collection and analysis, decision to publish, or preparation of the manuscript.

**Competing interests:** Novartis Pharmaceuticals Corporation provided the research group with Sacubitril/valsartan and valsartan used in this study free of charge. Dr. Hussain was previously employed by Novartis Pharmaceuticals Corporation at the time of study conduct, but he was no longer an employee of the company when the manuscript was draft, and it was voluntary for the study group to adjust the manuscript according to his comments. There were no restrictions or involvement from Novartis Pharmaceuticals Corporation related to the study design, data collection and analysis, decision to publish, or preparation of the manuscript. This does not alter our adherence to PLOS ONE policies on sharing data and materials.

## Introduction

Ischemic heart disease is the most common cause of heart failure (HF), and acute myocardial infarction (MI) can lead to acute and chronic HF [1–3]. Furthermore, HF can cause renal dysfunction and damage, a condition known as cardiorenal syndrome (CRS), increasing all-cause mortality in patients with HF [4,5] and limiting therapeutic options in HF treatment [6,7].

Sacubitril/valsartan (Sac/Val) is the first-in-class combined angiotensin receptor blocker and neprilysin inhibitor (ARNI) that has been approved for the treatment of HF with reduced ejection fraction, and it is recommended as first line treatment in the 2022 American guidelines for HF management [8]. However, concerns have been raised regarding its renal effects [9].

Sacubitril is a neprilysin inhibitor which prevents the clearance of several endogenous vasoactive peptides, ultimately leading to natriuresis, diuresis and vasodilatation [10]. However, neprilysin inhibitors also increase levels of endogenous vasoconstrictors like angiotensin II, resulting in the need for simultaneous inhibition of the renin angiotensin aldosterone system (RAAS) to ensure beneficial cardiovascular effects [10]. The net vasodilatory, natriuretic, and diuretic effects of combined sacubitril and valsartan treatment likely contributes to lower blood pressure (BP) levels beyond the BP reduction observed with monotherapy with RAAS inhibitors (i.e., angiotensin receptor blocker or angiotensin-converting enzyme inhibitor) [11, 12]. Concerns regarding hypotension and worsening renal function are important reasons for the under-use and discontinuation of RAAS inhibitors alone [7]. Physicians will have concerns with the use of ARNIs in hemodynamically unstable conditions like acute MI due to the risk of excessive BP lowering that may exceed the renal capacity for autoregulation of glomerular perfusion pressure [13].

Meta-analyses have demonstrated fewer renal adverse events in HF patients treated with ARNI compared to RAAS inhibition, especially in HF with preserved ejection fraction [14, 15]. While some clinical studies have not reported creatinine changes or have excluded patients with severe renal dysfunction [16–19], the renal effects of early ARNI initialization after an acute MI have not been thoroughly investigated. Sac/Val treatment effect on overall kidney function in the post-MI setting remains an important research question.

We aimed to investigate the effects of Sac/Val on kidney function and structure in a rodent model of combined cardiac and renal dysfunction after MI.

## Materials and methods

The study was conducted according to the Norwegian Animal Welfare Act, which conforms with the "European Directive 2010/63/EU on the protection of animals used for scientific purposes" and was approved by the Norwegian Food Safety Authority (FOTS approval #10102).

### Animal model and study design

Our highly experienced surgical team performed left anterior descending artery ligation to induce MI in male Sprague-Dawley rats weighing approximately 250 g (range

187–323 g, about 10 weeks old) on day 0. Anesthesia was induced with 2.5% isoflurane and 97.5% $O_2$, and anesthesia was maintained with 2–3% isoflurane. The animals were intubated and ventilated on a small-animal ventilator (VentElite, Harvard Apparatus, MA). After a left-sided thoracotomy and pericardiotomy the left anterior descending artery was ligated with a 7.0 non-absorbable thread without exteriorizing the heart. If there was insufficient blanching and hypokinesia of the myocardium, a second suture was applied. The thoracotomy was closed and the animal extubated and observed for signs of complications. All rats were kept warm using an electrical heating plate until they returned to normal activity. Sham animals underwent the same procedure except for the coronary artery ligation [20]. Subcutaneous buprenorphine at 0.05 mg/kg was used for preoperative analgesia 30 minutes before the procedure, and for postoperative analgesia in repeated doses.

Animals with MI size ≥35% of the left ventricular (LV) area based on magnetic resonance imaging (MRI) taken the day after operation were randomly allocated to oral treatment with either vehicle, valsartan or Sac/Val. Only animals logistically available for collection of renal parameters, i.e., blood pressure measurement, urine collection and/or serum for creatinine measurement were included in this renal study. 1 animal in the Sac/Val group was retrospectively excluded as it had an atrophic right kidney found during harvesting. S1 Fig shows that the number of animals available for inclusion in each group was n = 20 for vehicle, n = 14 for valsartan and n = 15 for Sac/Val. A group of sham operated animals was also included (n = 25). We included a higher number of sham/vehicle animals because they were also used in other, parallel investigations. Our previously published article on cardiac effects included the same animals as in this study in addition to an extended subset of animals for analysis of left ventricular ejection fraction, left ventricular end-diastolic volume and left atrial diameter [21].

Fig 1 shows the timeline of the study. Treatment started on the second day after the MI/sham procedure and lasted for six weeks. Drugs were dissolved in distilled water and administered at a dose of 31 mg/kg/day for valsartan and 68 mg/kg/day for Sac/Val. All animals received 4 mL/kg solution by oral gavage daily, with the treatment groups receiving valsartan or Sac/Val, and the sham and vehicle animals receiving distilled water only. The total dose received were adjusted according to weekly weighing. Oral gavage was only performed by a limited number of experienced staff to reduce distress in the animals related to the procedure.

The animals were housed two to three animals per cage, except for the 48 hours of isolation in metabolic cages. They were kept in a room with a 12-hour light/dark cycle and with set temperature and humidity. They had free access to food and water. The animals were monitored daily, with two inspections per day for the first three days postoperatively. All staff handling the animals had completed a certificate course in laboratory animal science (FELASA).

Several criteria were defined as humane end points. Signs of decompensated heart failure like reduced spontaneous activity combined with piloerection, stridorous breathing, peripheral edemas especially around the eye lids, and potentially frothing around nose/mouth. Postoperative wound infection with larger amounts of pus or suspicion of infection with

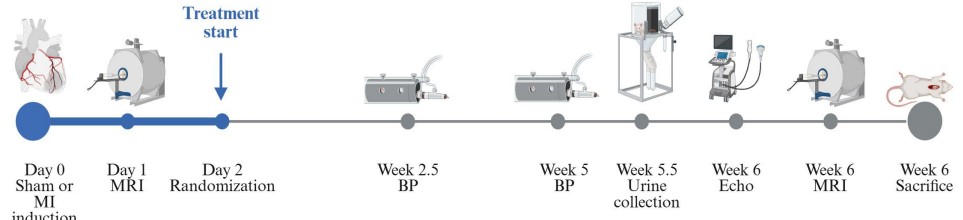

**Fig 1. Study timeline.** *Note.* BP = Blood pressure, Echo = Echocardiogram, MI = myocardial infarction, MRI = Magnetic resonance imaging. Created in BioRender. Bergo, **K.** (2026) https://BioRender.com/xdmzx0o.

reduced general condition like reduced activity, vocalizing, averting behavior when touched around the wound, inflammatory signs around the wound. Larger openings of the operation wounds over approximately 1x1 cm or suspected severe complications from oral gavage like significant change of respiration or behavior during or shortly after the oral gavage procedure. No animals reached humane end points before sacrifice. 3 animals included in the analyses in the Sac/Val group were either found dead in their cage or died during MRI after 5.5–6.0 weeks. The animals found dead were not registered with signs of humane end points in advance, and it is suspected that they died of sudden cardiac arrythmia or acutely decompensated heart failure due to the cardiodepressive effects of isoflurane anesthesia used during MRI.

## Procedures

### Noninvasive arterial blood pressure (BP) measurements

Tail-cuff measurement of BP was performed in conscious rats using the volume-pressure-recording technique at two and a half and five weeks after randomization using the CODA High Throughput System (Kent Scientific Corp., Torrington, CT, USA) [22]. The animals were constrained in specialized containers, placed on a warming platform and covered with a dark blanket during the experiment. Five acclimation cycles were performed, before 40 consecutive measurements were taken, and the average of accepted BP measurements was registered (automatic, pre-specified criteria by the CODA System).

### 24-hour urine collection, urine and serum creatinine sampling

Animals were placed in metabolic cages (Tecniplast, Buguggiate, Italy) and given 16−24 hours for acclimatization before the 24-hour urine collection started. During collection, the urine was cooled <10°C, a urine sample was then taken from the 24-hour urine collection and stored at −80°C for urinary analyses of sodium, kidney injury molecule-1 (KIM-1), neutrophil gelatinase-associated lipocalin (NGAL), creatinine and albumin. The animals had free access to food and water, and the collection took place after the noninvasive BP measurements at five and a half weeks. Serum creatinine was measured in venous blood samples taken from the vena saphena lateralis immediately after the urine collection ended.

### Echocardiographic assessment

Echocardiography was performed before sacrifice at six weeks, on animals anesthetized with 1.75% isoflurane and 98.25% $O_2$. The rats were kept warm on an electrical heating plate that allowed continuous electrocardiogram recordings during the examination. M-mode measurements of the LV and left atrium were performed in the long axis view, together with pulsed-wave Doppler measurements of posterior wall tissue velocity. Pulsed-wave Doppler was also used for mitral flow measurement. A single, blinded operator analyzed the echocardiographic data, using Vevo-3100 software (Fujifilm Visualsonics, Canada).

### Magnetic resonance imaging (MRI)

We performed MRI using a 9.4T MR system (Agilent Technologies Inc., Santa Clara, CA, USA) with dedicated hardware for rodents on day 1 to quantify MI size based on late gadolinium enhancement datasets, using Segment v3.0 (Medviso, segment.heiberg.se) [23]. MRI was repeated before sacrifice at six weeks for assessment of cardiac size and function. The animals were anesthetized with 1.75% isoflurane and 98.25% $O_2$, with small adjustments to keep heart rate and respiratory frequency stable, and heated air was used to maintain body temperature around 37 ˚C.

   At six weeks LV end-systolic and end-diastolic volumes were analyzed from a stack of short-axis CINE images by delineation of the endocardium and epicardium in all slices that contained LV myocardium, also using Segment v3.0 (Medviso, segment.heiberg.se) [24]. Ejection fraction was derived from the volume measurements.

## Sacrifice at six weeks

A subset of the animals underwent LV catheterization from the right carotid artery just prior to the harvesting procedure (n = 24 for sham, 20 for vehicle, 13 for valsartan and 11 for Sac/Val), results published previously [21]. After cardiac catheterization, heparin and EDTA plasma samples were taken from the vena jugularis externa or vena cava inferior for aldosterone and angiotensin II analyses (heparinized plasma), and for analysis of atrial natriuretic peptide (ANP) (EDTA plasma).

The animals were then placed in deep isoflurane anesthesia and the distal abdominal aorta was cannulated and ligated proximal and distal to the renal arteries before the kidneys were perfused with ice-cold PBS [25]. In a subset of the animals, a clamp was placed on the right renal artery, so that the left kidney was perfused while leaving the right kidney unperfused for weighing (n = 16 for sham, 15 for vehicle, 10 for valsartan and 7 for Sac/Val). The kidneys were removed, decapsulated and weighed. The best perfused kidney was cut in a transversal slice that was fixed in 4% buffered formaldehyde for histological analyses. The lungs were excised and weighed, and the right tibia was excised and measured.

## Blood and urine biochemical analyses

Sodium and creatinine analyses were performed on a COBAS 8000 c702 (Roche Diagnostics, Mannheim, Germany) at the Department of Medical Biochemistry at Oslo University Hospital. Serum and urine creatinine were determined by enzymatic colorimetric method, while urine sodium was analyzed with the indirect ion-selective electrodes method.

## Enzyme-linked immunosorbent assay (ELISA)

Commercially available ELISA kits with pre-coated plates were used according to manufacturer's instructions. Sigmaplot (version 14.0, Systat Software Inc, California, USA) was used for 4-parameter logistic curve fitting. For samples with values outside the detection range of the kit, the corresponding lowest or highest standard value was noted as that sample's result. Urinary analytes were normalized to creatinine. Samples were analyzed in duplicate and averaged for each animal. To reduce the significance of batch effects between kits, results from samples that were run on separate plates for the same analyte were normalized to the arithmetic mean of samples from two healthy animals, also run in duplicates, that were analyzed on all plates. We also made sure that samples from each group were represented on each plate. The following assays were used: Aldosterone (R&D, Minneapolis, MN, USA; cat.no: KGE016), angiotensin II (Abnova, Taipei, Taiwan; cat.no: KA1679), ANP (RayBiotech, Norcross, GA, USA; cat.no: EIA-ANP), albumin (Novus Biologicals, Abingdon, UK; NBP2−60525), KIM-1 (R&D, Minneapolis, MN, USA; cat.no: RKM100) and NGAL (BioPorto Diagnostics, Hellerup, DK; cat.no: 046).

## Histological analyses

Six representative formalin-fixed paraffin-embedded kidneys from each group were prepared, and sections of thickness three μm were stained with periodic acid schiff (PAS), following standard procedure at the Department of Pathology, Oslo University Hospital. Sections of thickness four μm were stained with picrosirius red according to the manufacturer's protocol (Polysciences, Warrington, PA, USA). Stained slides were scanned at x20 bright field magnification with AxioScan Z1 (Zeiss, Jena, Germany).

To evaluate glomerular size in a random selection of glomeruli, PAS sections were divided in outer and inner cortex. Then a macro run on Matlab R2019b (Mathworks, Natick, MA, USA) divided the inner cortex into 100 quadrants and the outer cortex in 144 quadrants and performed an automated, random selection of the quadrants. ImageJ (version 1.52j) was used to measure the area of the glomerular capillary convolute until at least 16 whole glomeruli of the inner cortex and at least 24 whole glomeruli of the outer cortex were measured. The glomerular areas were averaged for each animal before statistical analysis.

Collagen quantification was performed on picrosirius red sections with ZEN2 software, blue edition (Zeiss, Jena, Germany), applying custom thresholds for separation of stained areas.

Interstitial collagen deposition in the cortex and outer stripe of the outer medulla was quantified as the collagen-stained area divided by total tissue area, excluding the renal capsule, large blood vessels and artefacts. To investigate perivascular fibrosis, all visible arteries and arterioles were evaluated, and only arteries with a form factor of >0.75 were included in the analysis (calculated with the formula $4\pi \times (vessel\ area)/(vessel\ perimeter^2)$, the form factor of a perfect circle is 1) [26]. The ratio of the perivascular collagen-stained area to the total vessel area was averaged for all arteries taken from the same section. All histological analyses were performed by a single operator blinded to treatment group.

## Statistical methods

The power analysis of the major study [21] was based on an expected change of EF with 15%, standard deviation (SD) 0.2, with power 0.8 and α = 0.05, leading to the need of 22 animals in each group. This was based on our group's previous experience with the model. We did not perform a separate power analysis of this renal substudy. The attrition of animals caused a reduction in power, and non-significant results must be interpreted with care.

For normality assessment, a combination of visual methods (inspection of histograms and normal Q-Q plots) and Shapiro Wilk normality test was used. Comparisons between all groups were done with one-way ANOVA with bootstrap followed by Tukey's post hoc test. When the data did not follow normal distribution, Kruskal-Wallis test with subsequent Dunn's test was used. Comparison between two groups (i.e., kidney weight in sham and MI animals) was performed with independent samples t-test with bootstrap. A two-sided α level of <.05 was considered significant. Missing values were excluded case-wise. All statistical analyses were performed with IBM SPSS Statistics 29.0. Data are presented as mean and SD or median and interquartile range [IQR], as specified for each figure or table.

## Results

### Our MI rodent model developed systolic dysfunction without reduction in BP levels

Infarction size, baseline body weight, body weight at study end and tibia length did not differ significantly between groups (Table 1). Vehicle treated rats exhibited cardiac dysfunction with reduced ejection fraction and LV dilatation at week 6 (Table 2). There was a tendency for left atrial enlargement (p = 0.062 vs sham), but despite reduced systolic function almost all animals had no signs of pulmonary congestion as indicated by lung weight (Table 2). BP levels were not significantly different compared to sham at six weeks after MI (Fig 2, Table 2). The heart rate was slightly lower in MI compared to sham after two and a half weeks, but not significantly different after five weeks.

**Table 1. Animal characteristics.**

| Variable | Sham | Vehicle | Valsartan | Sac/Val | Overall p-value |
|---|---|---|---|---|---|
| **Infarct size (%)** | – | 45.7 [41.2-54.8] | 47.3 [39.7-52.1] | 43.9 [38.7-47.5] | 0.541 |
| **Body weight at baseline (g)** | 252 [230-280] | 246 [236-266] | 237 [215-275] | 248 [225-259] | 0.515 |
| **Body weight at sacrifice (g)** | 445±38 | 431±38 | 415±31 | 426±26 | 0.070 |
| **Tibia length at sacrifice (mm)** | 41.3 [40.8-42.0] | 41.0 [40.4-41.7] | 40.6 [40.0-41.5] | 40.3 [39.9-41.0] | 0.099 |

*Note.* Infarct size, n = Vehicle: 20, Valsartan: 14, Sac/Val: 15. Baseline body weight, n = Sham: 23, Vehicle: 20, Valsartan: 14, Sac/Val: 15. Body weight at sacrifice, n = Sham: 25, Vehicle: 20, Valsartan: 14, Sac/Val: 13. Tibia length, n = Sham: 25, Vehicle: 20, Valsartan: 14, Sac/Val: 12. Parameters presented as median [interquartile range], except infarct size presented as mean ± SD. Kruskal Wallis test performed on all parameters except body weight at sacrifice, where one-way Analysis of Variance (ANOVA) was done.

Table 2. Cardiovascular function at six weeks.

| Variable | Sham | Vehicle | Valsartan | Sac/Val | Overall p-value |
|---|---|---|---|---|---|
| LVEF (%) | 70.2 ± 5.1 | 42.6 *** ± 5.7 | 45.3 *** ± 6.2 | 45.6 *** ± 6.8 | < 0.001 |
| Fractional shortening | 0.37 [0.34-0.43] | 0.16 *** [0.14-0.19] | 0.17 *** [0.15-0.21] | 0.17 *** [0.14-0.21] | < 0.001 |
| LVEDV (mL) | 0.66 [0.62-0.74] | 1.10 *** [0.92-1.32] | 1.06 *** [0.85-1.18] | 0.87 *** [0.84-1.09] | < 0.001 |
| LVIDd (mm) | 7.9 [7.6-8.4] | 10.2 *** [9.7-11.2] | 10.4 *** [9.5-11.5] | 10.0 *** [9.0-10.3] | < 0.001 |
| Left atrial diameter (mm) | 4.1 [3.7-4.4] | 4.6 [3.8-5.6] | 4.6 * [4.1-5.5] | 3.9 †, § [3.6-4.5] | 0.025 |
| Lung weight (g) | 1.44 [1.34-1.55] | 1.43 [1.37-1.61] | 1.48 [1.33-1.57] | 1.39 [1.30-1.45] | 0.351 |
| SBP at 2.5 weeks (mmHg) | 158 [146-174] | 147 [133-162] | 134 *** [114-147] | 144 * [132-153] | 0.002 |
| SBP at five weeks (mmHg) | 166 [148-172] | 152 [143-160] | 136 ***, † [126-142] | 138 ***, † [125-149] | < 0.001 |
| DBP at 2.5 weeks (mmHg) | 112 [100-122] | 103 [92-115] | 91 ** [79-103] | 101 [92-114] | 0.022 |
| DBP at five weeks (mmHg) | 113 [101-123] | 109 [101-118] | 91 ***, †† [86-98] | 98 ** [87-103] | < 0.001 |

*Note.* Left ventricular ejection fraction (LVEF) and end-diastolic volume (LVEDV) are MRI derived, n = Sham: 20, Vehicle: 19, Valsartan: 6, Sac/Val: 10. Fractional shortening, left ventricular internal diameter end-diastole (LVIDd) and left atrial diameter from echocardiography, n = Sham: 25, Vehicle: 20, Valsartan: 13, Sac/Val: 11. Lung weight, n = Sham: 25, Vehicle: 20, Valsartan: 14, Sac/Val: 12. Blood pressure measurements, n = Sham: 22, Vehicle: 16, Valsartan: 12, Sac/Val: 12. Parameters presented as median [interquartile range], except LVEF presented as mean ± SD. Kruskal Wallis with subsequent Dunn's post hoc test performed on all parameters except LVEF, where one-way Analysis of Variance (ANOVA) with Tukey post hoc test was done.

*denotes p < 0.05 vs sham, **denotes p < 0.01 vs sham, ***denotes p < 0.001 vs sham.

†denotes p < 0.05 vs vehicle, ††denotes p < 0.01 vs vehicle.

§denotes p < 0.05 vs valsartan.

DBP = Diastolic blood pressure, MRI = Magnetic resonance imaging, SBP = Systolic blood pressure.

## MI animals developed renal dysfunction with no signs of structural renal damage

The vehicle treated MI animals developed mild renal dysfunction after five and a half weeks, as demonstrated by increased serum creatinine and reduced creatinine clearance compared to sham (Fig 3). Due to possible differences in fluid retention between the groups, we evaluated creatinine clearance without normalization to body weight. However, normalization of creatinine clearance to body weight or tibia length reflecting body size did not change significance levels between the animal groups (S1-S2 Fig). The animals' natriuresis and diuresis were not significantly different (Fig 3). Compared to the sham group, the vehicle treated group did not exhibit higher interstitial or perivascular collagen deposition in the kidneys, nor did they have significantly different urinary renal damage markers like albuminuria, KIM-1 or NGAL (Fig 3 and 4). No overt histological changes were detected on PAS or picrosirius red staining of the kidneys although the quality of some samples was suboptimal (Fig 4). Kidney weight and glomerular size in the vehicle group were unchanged compared to sham animals (Fig 4).

## Systemic RAAS and ANP levels were unchanged in untreated MI animals

Six weeks after MI, RAAS activation in vehicle treated animals was not significantly different compared to sham, as measured by circulating aldosterone and angiotensin II, and both plasma ANP levels and urinary ANP excretion were unchanged (Table 3).

**Drug treatment reduced BP levels and Sac/Val reduced left atrial diameter**

Neither valsartan nor Sac/Val treatment improved cardiac systolic function nor LV dilatation after MI, however, only Sac/Val reduced left atrial diameter (Table 2). BP levels were reduced in both treatment groups compared to sham and vehicle towards study end, but halfway through the study both systolic and diastolic BP levels were reduced in the valsartan group while Sac/Val only reduced the systolic BP (Fig 2, Table 2). All MI groups had slightly lower heart rates compared to sham after two and a half weeks (Fig 2).

Our previously published report on a slightly larger subset of the animals included in the major trial, did however, demonstrate reduced left ventricular end-diastolic volume in the Sac/Val group compared to vehicle treatment [21]. The current investigation is probably underpowered to demonstrate such an improvement in cardiac function.

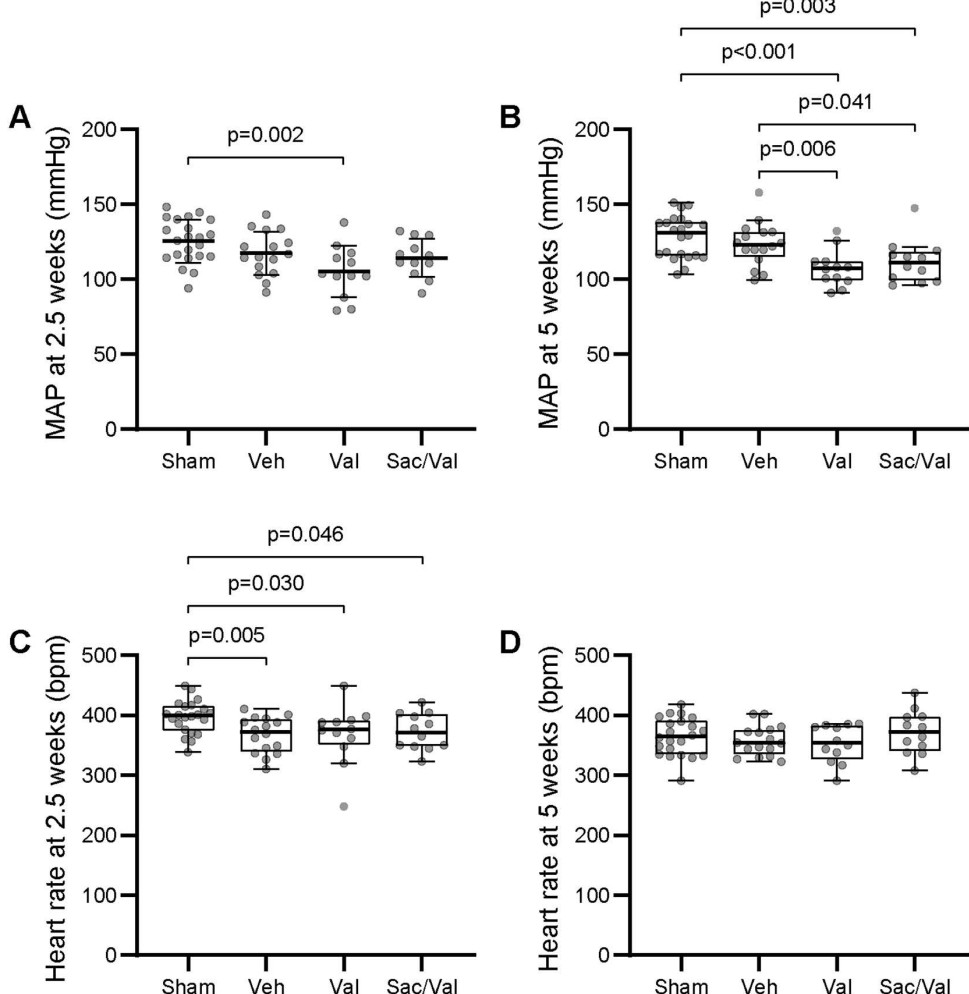

**Fig 2. Blood pressure levels and heart rates at two and a half and five weeks.** *Note.* Mean arterial pressure (MAP) at **Panel A**: Two and a half weeks (mean±SD, one-way Analysis of Variance (ANOVA) with Tukey post hoc test, overall p=0.003), and at **panel B:** Five weeks (median and IQR, Kruskal Wallis with subsequent Dunn's test, overall p<0.001). Heart rate at **Panel C and D:** Two and a half and five weeks (median and IQR, Kruskal Wallis with subsequent Dunn's test, overall p=0.021 at two and a half weeks, and not significant at five weeks). Bpm=Beats per minute, Sac/Val=Sacubitril/valsartan, Val=Valsartan, Veh=Vehicle.

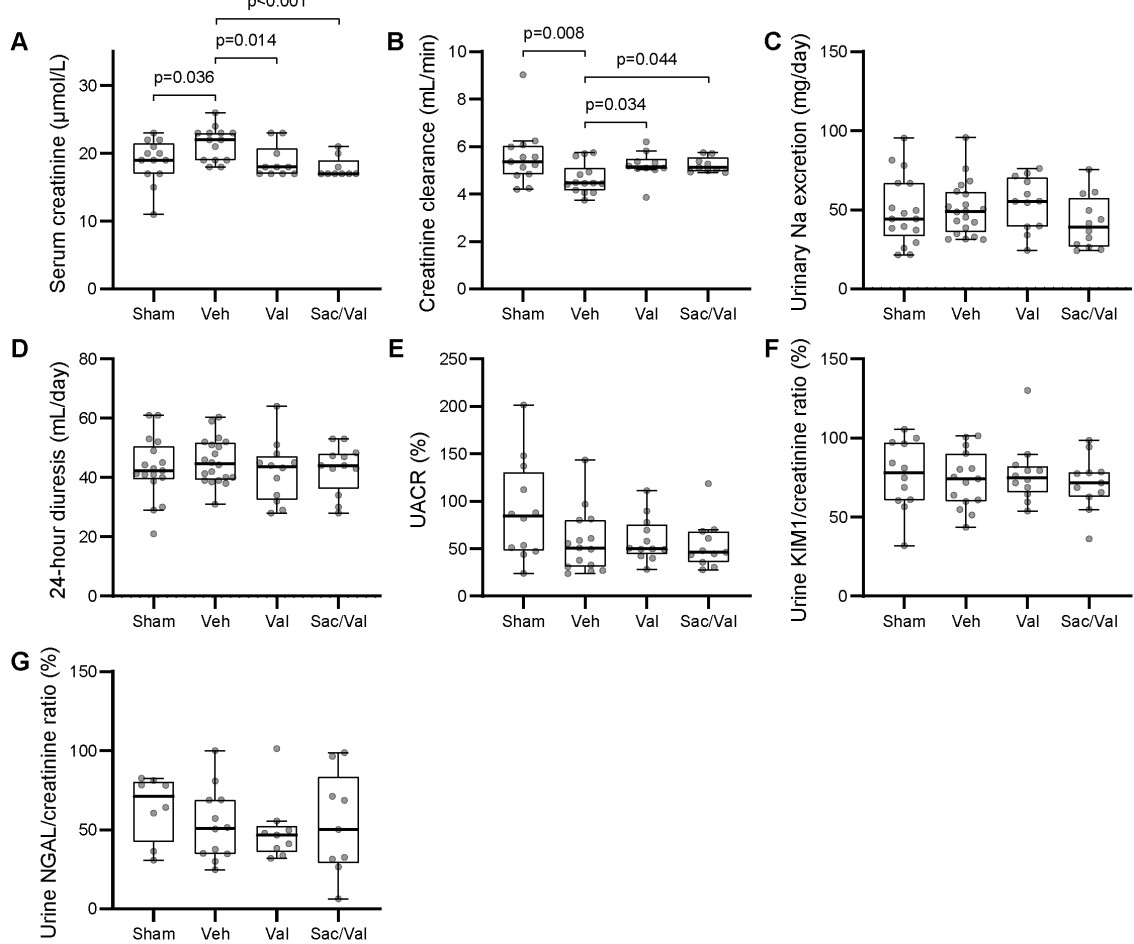

**Fig 3. Renal function and urinary markers.** *Note.* Functional renal parameters at ***Panels A-D:*** Five weeks (median and IQR, Kruskal Wallis with subsequent Dunn's test). Overall p = 0.006 for serum creatinine, p = 0.034 for creatinine clearance, and not significant for both urinary sodium excretion and urine volume. Urinary damage markers measured by ELISA in ***Panels E-G:*** Five weeks (median and IQR, Kruskal Wallis with subsequent Dunn's test). Samples were normalized to the arithmetic mean of the same control samples from two healthy animals analyzed on each ELISA plate, hence the unit of measurement is % of control samples. ELISA = Enzyme-linked immunosorbent assay, KIM-1 = Kidney injury molecule 1, Na = sodium, NGAL = Neutrophil gelatinase-associated lipocalin, Sac/Val = Sacubitril/valsartan, Val = Valsartan, Veh = Vehicle.

### Drug treatment preserved renal function despite lower BP levels

Both valsartan and Sac/Val prevented the observed reduction of renal function (Fig 3A-3B, supplemental S2 Fig). This renal effect occurred despite reduced BP levels (Fig 2, Table 2). Neither valsartan nor Sac/Val changed the urinary sodium excretion nor the 24-hour urine volume beyond physiological levels (Fig 3C-3D).

  Although the vehicle group did not have significantly lower kidney weight than the sham group, the valsartan animals did (Fig 4C). The observed reduced kidney weight was also observed when all the MI animals were compared to the sham animals (sham animals mean weight 1.34 ± 0.16 g and MI animals 1.22 ± 0.11 g, p = 0.020).

### Circulating RAAS and ANP levels were higher in the Sac/Val group

Circulating levels of angiotensin II were elevated in the Sac/Val group compared with all other groups, and plasma ANP levels were higher in the Sac/Val group compared to valsartan (Table 3). Systemic aldosterone and urinary ANP levels were not significantly different with treatment (Table 3).

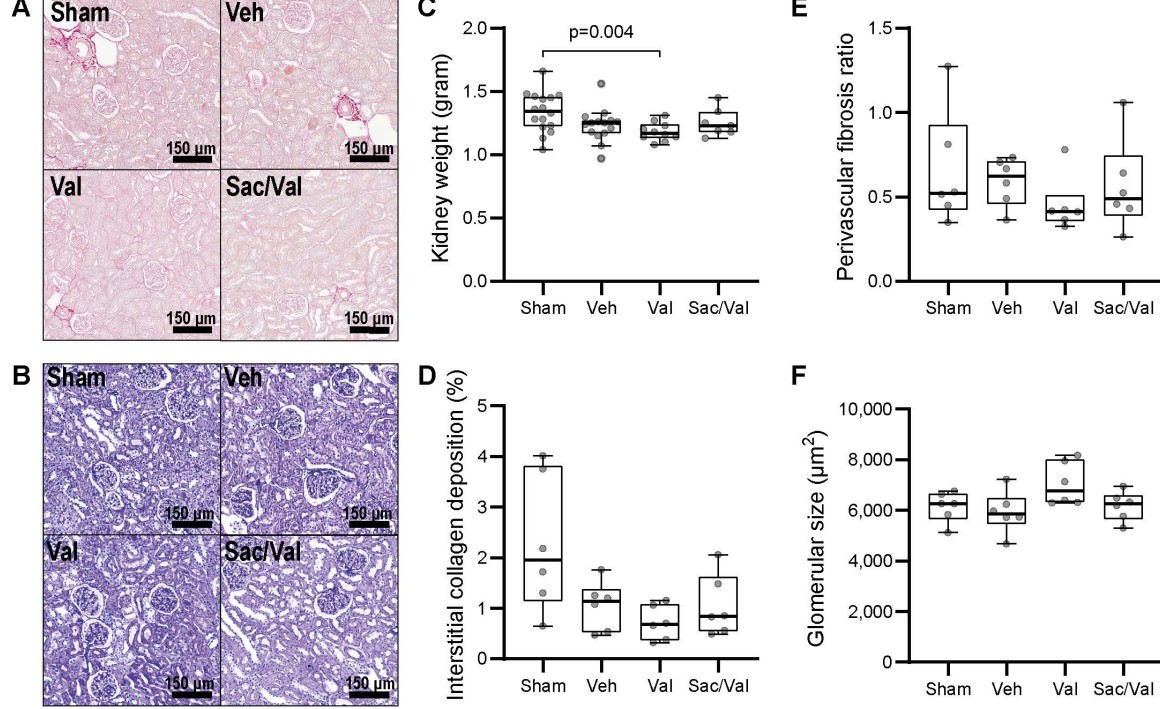

**Fig 4. Renal characteristics.** *Note.* Representative images of renal cortical sections with **Panel A:** Picrosirius red staining, and **Panel B:** PAS staining. Adjustment of contrast to match reference image have been applied in all images. Structural kidney parameters in **Panels C-F:** Six weeks (median and IQR, Kruskal Wallis with subsequent Dunn's test). Overall p = 0.032 for kidney weight, and not significant for the rest of the parameters. PAS = periodic acid schiff, Sac/Val = Sacubitril/valsartan, Val = Valsartan, Veh = Vehicle.

**Table 3. Hormonal markers of drug activity.**

| Variable | Sham | Vehicle | Valsartan | Sac/Val | Overall p-value |
|---|---|---|---|---|---|
| **Plasma ANP (%)** | 69.8 [51.4-77.0] | 68.4 [52.2-77.8] | 56.9 [42.3-67.7] | 93.0§§ [65.2-103.9] | 0.046 |
| **Urinary ANP/creatinine ratio (%)** | 260 [46-743] | 459 [120-732] | 195 [51-927] | 57 [19-456] | 0.164 |
| **Plasma angiotensin II (%)** | 69.8 [53.1-91.5] | 78.8 [37.8-116.1] | 85.0 [62.0-128.2] | 256.2***, ††, § [93.3-362.0] | 0.005 |
| **Plasma aldosterone (%)** | 146 [112-230] | 182 [105-237] | 137 [64-169] | 140 [105-177] | 0.310 |

*Note.* Parameters presented as median [interquartile range], all analyzed with Kruskal Wallis with subsequent Dunn's post hoc test. Samples were normalized to the arithmetic mean of the same control samples from two healthy animals analyzed on each ELISA plate, hence the unit of measurement is % of control samples. Plasma ANP n = Sham: 19, Vehicle: 20, Valsartan: 14, Sac/Val: 12. Urinary ANP/creatinine ratio n = Sham: 12, Vehicle: 15, Valsartan: 12, Sac/Val: 11. Plasma angiotensin II and aldosterone n = Sham: 14, Vehicle: 20, Valsartan: 13, Sac/Val: 12*** denotes $p < 0.001$ vs sham.

††denotes $p < 0.01$ vs vehicle.

§denotes $p < 0.05$ vs valsartan, §§denotes $p < 0.01$ vs valsartan.

ANP = atrial natriuretic peptide, ELISA = Enzyme-linked immunosorbent assay.

## Discussion

In our rodent model of mild CRS after MI, both valsartan and Sac/Val preserved renal function as measured by serum creatinine levels and creatinine clearance. These protective effects occurred despite reduced systemic BP levels in both treatment groups. Furthermore, neither valsartan nor Sac/Val improved ejection fraction, suggesting that both valsartan alone and combined Sac/Val treatment was inherently renoprotective.

### Blood pressure and cardiac effects

Interestingly, our preclinical study with Sac/Val initiated shortly after acute MI did not demonstrate BP reductions in the Sac/Val group beyond the lower BP levels of valsartan monotherapy. This is in contrast with the increased BP lowering effect of Sac/Val compared to an angiotensin converting enzyme inhibitor in the PARADISE-MI trial. Of note, valsartan has previously been found to lower BP levels more than an angiotensin converting enzyme inhibitor in the setting of acute MI [27]. This could indicate that after acute MI, Sac/Val may not excessively lower BP levels compared to valsartan alone.

The lower heart rate demonstrated early in all the MI groups could be due to remodeling of cardiac ion channels causing sinoatrial node dysfunction after MI [28].

Although global systolic dysfunction did not change with treatment in the current study, only the Sac/Val group reduced left atrial dilatation despite similar effects on BP and ejection fraction in the valsartan group. Increased natriuretic peptide levels may reduce both cardiac preload through their natriuretic and diuretic effects and afterload through vasodilatation [10], which could lead to the observed normalization of left atrial size.

### Renal function and structure

One possible mechanism through which Sac/Val may preserve renal function in the setting of lower systemic BP and RAAS inhibition as seen in the current study, is through elevated ANP levels. ANP is one of the substrates with highest affinity for neprilysin, and has been shown to cause vasodilatation of afferent arterioles and vasoconstriction of efferent arterioles, leading to increased intraglomerular pressure and increased glomerular filtration rate [29, 30]. However, the valsartan group improved creatinine clearance to the same extent as Sac/Val.

The natriuretic and diuretic effects of augmented natriuretic peptide levels could reduce fluid retention and lower central venous pressure [10]. This may further ameliorate the renal congestion associated with CRS originating from reduced cardiac function [5]. Although signs of obvious fluid retention, e.g., increased body weight and lung weight, were lacking in our study, mild to moderate fluid retention is harder to detect but could still be of pathophysiological relevance for the development of the CRS. Furthermore, renal effects were measured after five to six weeks in our study, by which time we expect the physiological changes associated with MI, such as acutely reduced cardiac systolic function, to have developed into a chronic phase. The natriuresis and diuresis five and a half weeks after an MI, with or without chronic drug treatment, may have reached a new state of equilibrium, which could explain why we did not see any differences between the groups in terms of sodium excretion and 24-hour urine volume.

The animals included in this study were young and still growing at the time of MI induction. The observed reduced kidney weight in MI animals is probably caused by lack of kidney growth during development due to sickness.

Cardiac dysfunction can cause renal dysfunction and damage through multiple pathophysiological pathways, such as neurohumoral and inflammatory alterations as well as the hemodynamic changes described above [5]. Enhancing the natriuretic peptide system can reduce renal inflammation and attenuate renal fibrosis [31], likely contributing to favorable renal effects of neprilysin inhibition. Although the rodent model of CRS after MI previously has been characterized by increased renal interstitial fibrosis and renal damage markers [32], our model was not associated with these alterations. This lack of structural renal damage despite functional decline might be due to the shorter observation time of our study compared to previous reports and is a unique model trait worth exploring further.

## Hormonal effects

Natriuretic peptides and the RAAS are critical regulators of cardiorenal homeostasis, significantly influencing fluid and electrolyte balance in opposing directions, and both hormonal systems are affected by Sac/Val [10].

The increased angiotensin II levels in the Sac/Val group were expected as neprilysin inhibits its degradation [10], but blood pressure levels are not elevated as valsartan inhibits its receptor. The circulatory RAAS levels in the vehicle group compared to sham were probably similar due to our model developing mild cardiac dysfunction without overt HF with pulmonary congestion. Circulating ANP levels were significantly increased in the Sac/Val group compared to valsartan treatment [9]. The lack of increased systemic ANP levels in the vehicle group compared to sham could be due to its short half-life [33]. It is also possible that local (i.e., cardiac, vascular and renal) ANP levels were elevated without detectable changes in the circulation.

## Limitations

The renal effects observed in this study are likely of hemodynamic origin and are hypothesis-generating analyses for future investigations. Limitations include lack of measured glomerular filtration rate, glomerular hemodynamic measurements, as well as central venous pressure measurements so that mechanisms behind the observed renal effects could be better understood.

## Conclusions

To our knowledge, this is the first preclinical study to investigate the early renal effect of Sac/Val in a rodent model transitioning from initial CRS type 1 following MI to CRS type 2 characterized by chronic cardiac dysfunction and renal impairment. Our study shows that Sac/Val treatment initiated in the acute phase after MI in rats preserves renal function. These effects seemed to be similar between Sac/Val and valsartan alone. Further preclinical and clinical trials are needed to corroborate these findings.

## Supporting information

**S1 Fig. Animal inclusion.** *Note.* 3 animals included in the analyses in the Sac/Val group were either found dead in their cage or died during MRI after 5.5–6.0 weeks. Created in BioRender. Bergo, K. (2026) https://BioRender.com/r84aurf.
(TIF)

**S2 Fig. Normalized creatinine clearance.** *Note.* Creatinine clearance normalized to **Panel A:** Body weight in grams and **Panel B:** tibia length at harvest in cm (median and IQR, Kruskal Wallis with subsequent Dunn's test). Overall p = 0.008 for creatinine clearance/body weight and p = 0.040 for creatinine clearance/tibia length. Sac/Val = Sacubitril/valsartan, Val = Valsartan, Veh = Vehicle.
(TIF)

## Acknowledgments

The authors thank Hilde Dishington, Dina Behmen, Almira Hasic, Hege Ugland, Tone Aksnes Lian, Marita Martinsen, Haelin Kim, Linn Espeland, Snorre Sanner Sjaastad, Andreas Romaine, Athiramol Sasi, Ioanni Veras, Cecilie Nome and Kari Løken for their valuable contribution to the study. The authors are grateful for the support from the KG Jebsen Center for Cardiac Research, University of Oslo, Oslo, Norway. Histological images were acquired at the Norbrain-Slidescanning Facility at the Institute of Basic Medical Sciences, University of Oslo, a resource funded by the Research Council of Norway

## Author contributions

**Conceptualization:** Kaja Knudsen Bergo, Emil Knut Stenersen Espe, Ivar Sjaastad, Alessandro Cataliotti.

**Data curation:** Kaja Knudsen Bergo, Ida Marie Hauge-Iversen, Lili Zhang.

**Formal analysis:** Kaja Knudsen Bergo, Einar Sjaastad Nordén, Ida Marie Hauge-Iversen, Lili Zhang.

**Funding acquisition:** Kaja Knudsen Bergo, Ivar Sjaastad, Alessandro Cataliotti.

**Investigation:** Kaja Knudsen Bergo, Einar Sjaastad Nordén, Bård Andre Bendiksen, Emil Knut Stenersen Espe, Gary McGinley, Ida Marie Hauge-Iversen, Lili Zhang, Ivar Sjaastad.

**Methodology:** Kaja Knudsen Bergo, Einar Sjaastad Nordén, Bård Andre Bendiksen, Emil Knut Stenersen Espe, Sabine Leh, Hans-Peter Marti, Lili Zhang, Ivar Sjaastad, Alessandro Cataliotti.

**Project administration:** Kaja Knudsen Bergo, Emil Knut Stenersen Espe, Ivar Sjaastad.

**Resources:** Kaja Knudsen Bergo, Einar Sjaastad Nordén, Bård Andre Bendiksen, Emil Knut Stenersen Espe, Gary McGinley, Ida Marie Hauge-Iversen, Rizwan Iqbal Hussain, Sabine Leh, Hans-Peter Marti, Lili Zhang, Ivar Sjaastad, Alessandro Cataliotti.

**Software:** Emil Knut Stenersen Espe, Ida Marie Hauge-Iversen.

**Supervision:** Kaja Knudsen Bergo, Ivar Sjaastad, Alessandro Cataliotti.

**Visualization:** Kaja Knudsen Bergo.

**Writing – original draft:** Kaja Knudsen Bergo.

**Writing – review & editing:** Kaja Knudsen Bergo, Einar Sjaastad Nordén, Bård Andre Bendiksen, Emil Knut Stenersen Espe, Gary McGinley, Ida Marie Hauge-Iversen, Rizwan Iqbal Hussain, Sabine Leh, Hans-Peter Marti, Lili Zhang, Ivar Sjaastad, Alessandro Cataliotti.

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
