## [Decision Letter · Decision Letter 0]

1 Jul 2025

Dear Dr. Bergo,

Thank you for submitting your manuscript to PLOS ONE. After careful consideration, we feel that it has merit but does not fully meet PLOS ONE’s publication criteria as it currently stands. Therefore, we invite you to submit a revised version of the manuscript that addresses the points raised during the review process.

We look forward to receiving your revised manuscript.

Kind regards,

Sepiso K. Masenga, PhD

Academic Editor

PLOS ONE

Journal Requirements: 

2. Funding Information and Financial Disclosure sections do not match:

We note that the grant information you provided in the ‘Funding Information’ and ‘Financial Disclosure’ sections do not match.

3.  Thank you for stating the following in the Competing Interests section;

 [Novartis Pharmaceuticals AG provided the research group with Sacubitril/valsartan and valsartan used in this study free of charge. Dr. Hussain was employed by Novartis Pharmaceuticals AG at the time of study conduct.]. 

We note that one or more of the authors have an affiliation to the commercial funders of this research study : [Novartis Pharmaceuticals AG].

Within your Competing Interests Statement, please confirm that this commercial affiliation does not alter your adherence to all PLOS ONE policies on sharing data and materials by including the following statement: ""This does not alter our adherence to  PLOS ONE policies on sharing data and materials.” (as detailed online in our guide for authors http://journals.plos.org/plosone/s/competing-interests). If this adherence statement is not accurate and  there are restrictions on sharing of data and/or materials, please state these. Please note that we cannot proceed with consideration of your article until this information has been declared.

Reviewers' comments:

Reviewer's Responses to Questions

**Comments to the Author**

1. Is the manuscript technically sound, and do the data support the conclusions?

Reviewer #1: Yes

Reviewer #2: Partly

2. Has the statistical analysis been performed appropriately and rigorously?

Reviewer #1: Yes

Reviewer #2: No

3. Have the authors made all data underlying the findings in their manuscript fully available?

Reviewer #1: Yes

Reviewer #2: No

4. Is the manuscript presented in an intelligible fashion and written in standard English?

Reviewer #1: Yes

Reviewer #2: Yes

Reviewer #1: Comment

The manuscript presents a clear experimental design, with a logical progression from rationale to methods and outcomes. Inclusion of relevant physiological parameters (creatinine clearance, MAP, cardiac imaging) provides a comprehensive picture of both renal and cardiac effects. The use of a post-MI rat model is clinically relevant and strengthens the translational value.

Points for Improvement

Abstract

The current title (“Sacubitril/valsartan preserves kidney function in rats after myocardial infarction”) does not fully capture the study's scope written in the abstract. Since the model involves post-MI systolic dysfunction (i.e., heart failure) and evaluates cardiorenal outcomes, the title should be revised to reflect this more accurately. Eg Sacubitril/valsartan preserves renal function in a rat model of post-MI heart failure or something like Cardiorenal protective effects of Sacubitril/valsartan after myocardial infarction in rats. Unlike just writing as after myocardial infarction

1. The abstract lacks essential terms which are included in the key words such as ARNI, Cardiorenal syndrome etc. Please ensure the key words are also appearing in your abstract

2. No mention of how Sacubitril/valsartan might exert renal protection (e.g., via RAAS-neprilysin modulation, natriuretic peptides). A single phrase would help anchor biological plausibility.

3. The concluding sentence states that Sac/Val and valsartan “preserve renal function to the same extent,” yet only Sac/Val reduced LA diameter, a meaningful difference that’s not emphasized in the conclusion. Refine the last sentence to reflect the additional cardiovascular benefit observed with Sac/Val.

4. P-values should be consistently formatted (e.g., p < 0.05, not p<.05), and consider standardizing data presentation (e.g., whether median [IQR] is needed throughout for all variables or just renal). Please include actual p-value unless it’s something like 0.000 of which you may justified to included a < sign.

Methods

The MI model is robust and well-described, and the sham group inclusion strengthens internal validity. Randomization, dosing regimens, animal handling, and humane endpoints are transparently outlined, showing high procedural quality.

Points for Improvement:

1. While deaths are explained in detail, consider briefly stating how attrition affected statistical power, and whether any imputation or exclusion strategy was pre-specified. For transparency, it would be helpful to include a flow diagram summarizing final numbers per group after exclusions.

2. The use of a fixed solution volume (4 mL/kg) is appropriate, but it's unclear if drug concentration adjustments were made for weight gain over the 6 weeks.

3. The note that some animals were used in another study [20] is appropriate. However, please clarify whether any overlapping endpoints or data reuse occurs in this manuscript.

4. The sequence of procedures (e.g., BP measurement at weeks 2.5 and 5, followed by urine collection at 5.5 weeks, then imaging at 6 weeks) is logical but would be clearer with a brief procedural timeline diagram or summary table.

5. It's still unclear if baseline renal and cardiac function (e.g., creatinine, EF) was confirmed to be normal before MI induction in all animals. This is essential for interpreting post-MI decline as treatment effects vs. pre-existing variability. were creatinine clearance values normalized to body surface area or body weight? This should be stated clearly. The use of reference samples from two healthy animals to normalize across plates is creative, but this approach should be justified and potential batch effects discussed.

6. The text refers to subsets undergoing LV catheterization and unilateral kidney perfusion, but the exact ‘n’ per procedure is not always stated clearly. This makes it harder to track sample sizes per outcome. It would be helpful to explicitly state how missing samples (e.g., deaths) were handled in analysis, was any imputation done, or were they excluded case-wise?

Statistical Methods:

1. The use of bootstrapped ANOVA and non-parametric tests is appropriate.

2. However, no mention is made of sample size justification or power calculation. Even a retrospective mention (e.g., based on effect sizes from primary endpoints) would strengthen confidence in findings.

3. Specify how multiple comparisons were addressed when testing more than one outcome (e.g., collagen, glomerular area, MAP, EF, etc.).

Results

1. The reduction in left atrial diameter with Sac/Val is important but under-discussed in the results. A more explicit acknowledgment that only Sac/Val achieved this structural improvement would strengthen interpretation, especially since systolic function was unchanged across treatments.

2. Again, While sample sizes are noted, it would be helpful to briefly mention how missing data from deaths or excluded MRI cases were handled in the final analysis…….especially since LVEF data are available in fewer animals than others.

3. The statement that "no visible histological changes were observed" is too casual, consider replacing with "no overt histological changes were detected on PAS or picrosirius red staining," and acknowledge the impact of suboptimal sample quality on interpretability.

4. The section heading claims RAAS and ANP levels were unchanged in MI animals, but Table 3 and text clearly show increased angiotensin II and ANP with Sac/Val. This contradiction needs correction for clarity.

5. The rise in angiotensin II with Sac/Val is expected due to reduced feedback inhibition, and elevated plasma ANP supports ARNI’s mechanism. However, the text does not clearly connect these findings to the observed clinical benefits (e.g., LA remodeling, renal protection). A sentence or two linking this more explicitly would improve cohesion.

6. The finding that valsartan reduced kidney weight is interesting but not explained. Was this due to hemodynamic unloading, structural remodeling, or measurement variation? A line of interpretation would help.

7. It is worth emphasizing that only Sac/Val improved LA diameter despite similar effects on BP and EF……..this supports a distinct hemodynamic or structural benefit and should be highlighted more assertively.

Discussion/conclusion/limitation

1. Refer to comment on LA diameter. Please reconcile or acknowledge this discrepancy to clarify whether Sac/Val confers any additive benefit.

2. The text attributes non-significant cardiac differences to being underpowered. It would help to quantify or cite effect sizes, or mention this limitation earlier under methods or results.

3. While plausible mechanisms for preserved renal function are discussed (e.g., afferent/efferent tone, ANP), the explanation is speculative. The discussion could benefit from highlighting the lack of structural damage despite functional decline as a unique model trait worth exploring further.

4. The limitations are appropriate but could be more sharply focused on missing functional readouts (e.g., no GFR, venous congestion), rather than repeating methods (light microscopy only).

5. E.g., "These protective effects occurred despite reduced systemic BP..." is restated multiple times across the text. Consider condensing to preserve flow and word economy.

Reviewer #2: Reviewer Comments:

1. Inclusion of Histological Evidence:

There is a need to include representative histological slides in the study to support the reported findings and enhance the interpretability of the results.

2. Abstract Revision:

The abstract should be revised to adhere to the journal’s guidelines. In particular, please ensure that the Methods section of the abstract includes key statistical details, such as the statistical tests used, and any software or analytical tools employed.

3. Clarification of Literature Context:

The sentence stating that "few clinical studies available did not report on creatinine changes or excluded patients with severe renal dysfunction" is broadly accurate. However, you may consider softening the language to reflect a more measured tone, such as: “While some clinical studies have not reported creatinine changes or have excluded patients with severe renal dysfunction...”

4. Paragraph Structure and Readability:

For improved readability, it is recommended that overly long paragraphs—particularly in the Animal Model and Study Design section—be split into shorter, logically structured paragraphs.

5. Sample Size Justification:

The manuscript does not clearly describe how the sample size was determined. Please include a justification or description of the sample size calculation or rationale, including any assumptions or references used.

**Do you want your identity to be public for this peer review?** For information about this choice, including consent withdrawal, please see our Privacy Policy

Reviewer #1: **Yes: ** Katongo Hope Mutengo

Reviewer #2: No

---

## [Author Response · Author response to Decision Letter 1]

2 Sep 2025

The specific reviewer and editor comments are addressed in the submitted Cover letter and rebuttal letter.

---

## [Decision Letter · Decision Letter 1]

8 Oct 2025

Sacubitril/valsartan preserves kidney function in rats with cardiorenal syndrome after myocardial infarction

PONE-D-25-26165R1

Dear Dr. Bergo,

We’re pleased to inform you that your manuscript has been judged scientifically suitable for publication and will be formally accepted for publication once it meets all outstanding technical requirements.

Kind regards,

Sepiso K. Masenga, PhD

Academic Editor

PLOS ONE

Additional Editor Comments (optional):

Reviewers' comments:

Reviewer's Responses to Questions

**Comments to the Author**

Reviewer #1: All comments have been addressed

Reviewer #2: All comments have been addressed

2. Is the manuscript technically sound, and do the data support the conclusions?

Reviewer #1: Yes

Reviewer #2: Yes

3. Has the statistical analysis been performed appropriately and rigorously?

Reviewer #1: Yes

Reviewer #2: Yes

4. Have the authors made all data underlying the findings in their manuscript fully available?

Reviewer #1: Yes

Reviewer #2: Yes

5. Is the manuscript presented in an intelligible fashion and written in standard English?

Reviewer #1: Yes

Reviewer #2: Yes

Reviewer #1: All reviews have been addressed. This is a well-conducted experimental study on sacubitril/valsartan in a post-MI rat model, with clear clinical relevance given the challenges of managing renal dysfunction in heart failure. The work is methodologically sound and provides new data on both renal and cardiac remodelling effects.

Reviewer #2: The manuscript investigates Sacubitril/valsartan (Sac/Val) in a rat model of myocardial infarction-induced cardiorenal syndrome. The study is well-designed, with clear randomization and assessment of renal and cardiac function using creatinine clearance, imaging, and MAP measurements. Both Sac/Val and valsartan preserved renal function compared to vehicle, while only Sac/Val reduced left atrial dilatation, suggesting additional cardiovascular benefits. Ethical approval and procedures are appropriate, with no concerns regarding dual publication or plagiarism

**Do you want your identity to be public for this peer review?** For information about this choice, including consent withdrawal, please see our Privacy Policy

Reviewer #1: No

Reviewer #2: No

---

## [Editor Report · Acceptance letter]

PONE-D-25-26165R1

PLOS ONE

Dear Dr. Bergo,

I'm pleased to inform you that your manuscript has been deemed suitable for publication in PLOS ONE. Congratulations! Your manuscript is now being handed over to our production team.

Kind regards,

on behalf of

Prof. Sepiso K. Masenga

Academic Editor

PLOS ONE